# Molecular characterization of xerosis cutis: A systematic review

Ruhul Amin [1,2], Anna Lechner[1], Annika Vogt[1], Ulrike Blume-Peytavi[1], Jan Kottner[3]*

1 Charité-Universitätsmedizin Berlin, Corporate Member of Freie Universität Berlin and Humboldt-Universität zu Berlin, Department of Dermatology, Venereology and Allergology, Clinical Research Center for Hair and Skin Science, Berlin, Germany, 2 BCSIR Laboratories Dhaka, Bangladesh Council of Scientific and Industrial Research, Dhaka, Bangladesh, 3 Charité-Universitätsmedizin Berlin, Institute of Clinical Nursing Science, Berlin, Germany

* jan.kottner@charite.de

## Abstract

### Background

Xerosis cutis or dry skin is a highly prevalent dermatological disorder especially in the elderly and in patients with underlying health conditions. In the past decades, numerous molecular markers have been investigated for their association with the occurrence or severity of skin dryness. The aim of this review was to summarize the molecular markers used in xerosis cutis research and to describe possible associations with different dry skin etiologies.

### Methods

We conducted a systematic review of molecular markers of xerosis cutis caused by internal or systemic changes. References published between 1990 and September 2020 were searched using 'MEDLINE', 'EMBASE' and 'Biological abstracts' databases. Study results were summarized and analyzed descriptively. The review protocol was registered in PROSPERO database (CRD42020214173).

### Results

A total of 21 study reports describing 72 molecules were identified including lipids, natural moisturizing factors (NMFs), proteins including cytokines and metabolites or metabolic products. Most frequently reported markers were ceramides, total free fatty acids, triglycerides and selected components of NMFs. Thirty-one markers were reported only once. Although, associations of these molecular markers with skin dryness were described, reports of unclear and/or no association were also frequent for nearly every marker.

### Conclusion

An unexpectedly high number of various molecules to quantify xerosis cutis was found. There is substantial heterogeneity regarding molecular marker selection, tissue sampling and laboratory analyses. Empirical evidence is also heterogeneous regarding possible associations with dry skin. Total free fatty acids, total ceramide, ceramide (NP), ceramide (NS), triglyceride, total free amino acids and serine seem to be relevant, but the association

**Funding:** This study was supported by Charité-Universitätsmedizin Berlin, Department of Dermatology, Venereology and Allergology, Clinical

Research Center for Hair and Skin Science, Germany. RA receives scholarship from 'Bangabandhu Science and Technology Fellowship trust, Ministry of Science and Technology, Bangladesh' to conduct his doctoral study in Charité-Universitätsmedizin Berlin. The funders had no role in study design, data collection and analysis, decision to publish, or preparation of this manuscript.

**Competing interests:** The authors have declared that no competing interests exist.

with dry skin is inconsistent. Although the quantification of molecular markers plays an important role in characterizing biological processes, pathogenic processes or pharmacologic responses, it is currently unclear which molecules work best in xerosis cutis.

## 1. Introduction

Xerosis cutis or asteatosis is caused by reduced hydration of the stratum corneum and characterized by clinical signs such as small to large scales, cracks, and inflammation [1]. This is often accompanied by pruritus and risks for secondary infections [2, 3]. Besides external causes and environmental triggers [4, 5], there are endogenous or intrinsic causes of xerosis cutis such as aging, internal health conditions, dermatological and psychiatric diseases, diet and drugs [6, 7]. For example, aging related physiological changes, hormonal alteration [8], disease induced stress and inflammatory response [9] or off-target activities of drugs [10] can affect skin hydration. Although the clinical signs and symptoms are similar, it can be assumed that, as different causes are involved, there are different underlying molecular mechanisms and pathways leading to xerosis cutis. In xerosis cutis, the stratum corneum (SC) fails to maintain an adequate water concentration gradient between the living epidermal cells and the skin surface [11]. The changes may also include a decreased sebum and sweat production, inadequate cell replacement [12], disturbed skin barrier function [1] and increased transepidermal water loss [13].

The SC consists of terminally differentiated and unnucleated keratinocytes, namely corneocytes, and a lipid matrix surrounding the cells [14]. The lipid matrix contains cholesterol, ceramides, fatty acids, cholesterol sulfate, glucosyl ceramides, phospholipids, proteins and enzymes [15–17]. Ceramides, which are essential for an optimal lipid structure, play an important role in determining water permeability and maintaining skin barrier function [15]. In addition, natural moisturizing factors (NMFs), mainly located in corneocytes [18], contribute to maintaining SC hydration [11]. Changes in the structure, arrangement or composition of any of these components may lead to decreased SC hydration and may affect the processes regulating skin integrity [43] and normal desquamation [32].

Today, biomarkers play important roles in clinical research and in dermatology. A biomarker is considered as "a characteristic that is objectively measured and evaluated as an indicator of normal biological processes, pathogenic processes or pharmacologic responses to a therapeutic intervention" [19]. From the early 1990's, there has been growing interest in molecular markers or compounds which are associated with the occurrence and/or the severity of skin dryness. Advances in analytical methods and instrumentations facilitated the laboratory analysis of molecules and the discovery of new markers [17, 20]. However, up to present time, diagnosis of xerosis cutis is largely based on clinical methods of visual assessment using scores or classifications [21, 22]. Whether the measurement of molecular markers is useful in dry skin assessment, is unclear. It may help to diagnose the underlying cause of xerosis cutis. In addition, changes of molecular markers may help to understand and/or to measure (early) treatment responses.

However, despite the wide range of markers used in xerosis cutis research [34, 37, 41, 43], there is no agreement yet about the most accurate and useful candidates. Therefore, the aim of this systematic review was to describe and summarize molecular markers of dry skin and to describe possible associations with clinical signs and/or the severity of xerosis cutis and possible underlying etiologies.

## 2. Methods

### 2.1. Eligibility criteria

We included primary studies in humans (all age groups and all languages) reporting quantitative data of molecular markers of dry skin along with performed analytical methodologies. Xerosis caused by intrinsic processes (e.g., due to aging) or underlying internal diseases (e.g. diabetes mellitus) was in our focus. The included studies had to include the participants' age, skin areas and symptoms and/or severity of dry skin. We excluded articles that described xerosis due to external causes, such as exposures to irritants, allergens, pathogens, topical treatments and inflammatory dermatological diseases such as dermatitis, psoriasis, eczema or comparable conditions. Reviews, letters, editorials, personal opinions, posters, conference abstracts as well as pre-clinical or animal studies and in vitro studies were not included in this review.

### 2.2. Information sources

'MEDLINE', 'EMBASE' and 'Biological Abstracts' databases were searched concurrently via OvidSP on 29 September 2020. We also conducted an updated database search on 1 January 2021 with exactly the same search criteria.

### 2.3. Search strategy

We searched the above-mentioned databases with combinations of key words covering xerosis cutis, humans and molecular markers. The search was conducted for articles published between 1990 and 29 September 2020. The reference lists of all interesting articles were also searched manually to identify any additional studies that fit the focus of our review. The detailed search strategy is presented in S1 Appendix.

### 2.4. Selection process

The retrieved titles and abstracts were independently screened by two reviewers (RA and AL) Any difference in opinions between the two reviewers was resolved by consensus or by the third reviewers (JK, AV). Full text articles of all potentially eligible studies were independently checked for eligibility by the reviewers (RA and AL) and then finalized by discussion with a third author.

### 2.5. Data collection process

From the included studies, two reviewers extracted data regarding main outcomes of the primary studies, details about study, study participants, intervention (if any) and quantification methods. A standardized data extraction form was used. If needed, quantities of molecular markers were extracted from graphs or figures. Study results were summarized descriptively.

### 2.6. Data items

The following items were extracted: author's name, publication year, study design, country/ethnicity, signs of dry skin and scoring method, analyzed material, sampling technique, method of analysis, number of participants, age, sex, skin areas, severity of dry skin, molecular markers, results and quantification units (S2 Appendix).

## 2.7. Risk of bias assessment

There are no accepted standards or methodological guidance how to best quantify molecular markers in skin research. In Addition, the objective of this review was to describe the occurrence and characteristics of the molecular markers. Therefore, a formal risk of bias assessment was not conducted.

## 2.8. Effect measures

Differences between groups and the degree and strength of associations were considered as effect measures.

## 2.9. Synthesis methods

Extracted study results were analyzed descriptively. In order to detect possible group differences, a simplified evaluation scheme was applied: differences between proportions or quantities of molecular markers between normal and dry skin of more than 10% were considered to indicate possible associations ('Yes, higher/ lower in dry skin'). Differences between 5% to 10% were considered unclear and indicated with a question mark (?). Any difference lower than 5% was considered as biological variation ('No').

When molecular markers were presented for at least three or more different dry skin severities, a consistent increase or decrease of the marker quantity with the corresponding category was considered as a possible association. One or two deviated values in the 'trend pattern' were considered as unclear association. If there were no differences among the markers' values in relation to different dry skin severities, an association was considered unlikely. A summary of possible association was made for all the markers presented in each included article. A list of top markers was prepared considering the numbers of studies reported the corresponding markers (at least two studies). Markers analyzed once were listed separately.

# 3. Results

## 3.1. Study selection

A total of 1858 records were yielded from electronic searches in 'Medline', 'Embase' and 'Biological Abstracts' databases via OvidSP. Based on title and abstract screening, 1675 records were excluded. The remaining 183 publications were retrieved for full text evaluation along with 13 more articles which were found while searching in reference lists. Out of these 196 references, 175 publications were excluded as they did not meet the inclusion criteria. Finally, 21 articles were included for data extraction [23–43] (Fig 1).

## 3.2. Study characteristics

Thirteen studies were designed as cross-sectional, four as randomized control trials, two as controlled clinical trials, two as case controls and the remaining one as pre-post study. Four studies were conducted in America, nine in Asia and eight in Europe. The sample size ranged from 13 to 159 and the age of the subjects ranged from 23 to 94 years. Two studies did not report the participant's age, six did not report participant's sex and six studies did not assessed the severity of dry skin using a classification or scoring method.

Different forms of xerosis cutis were investigated. Among the included articles, five examined elderly participants whose dry skin conditions were indicated either to be associated with aging [26] or as senile xerosis [25, 28, 33, 38] where especially older people had dry skin. Here, we represented this condition as 'senile xerosis'. Skin dryness of persons with diabetes is described as diabetic xerosis which may be considered as one particular form of xerosis cutis.

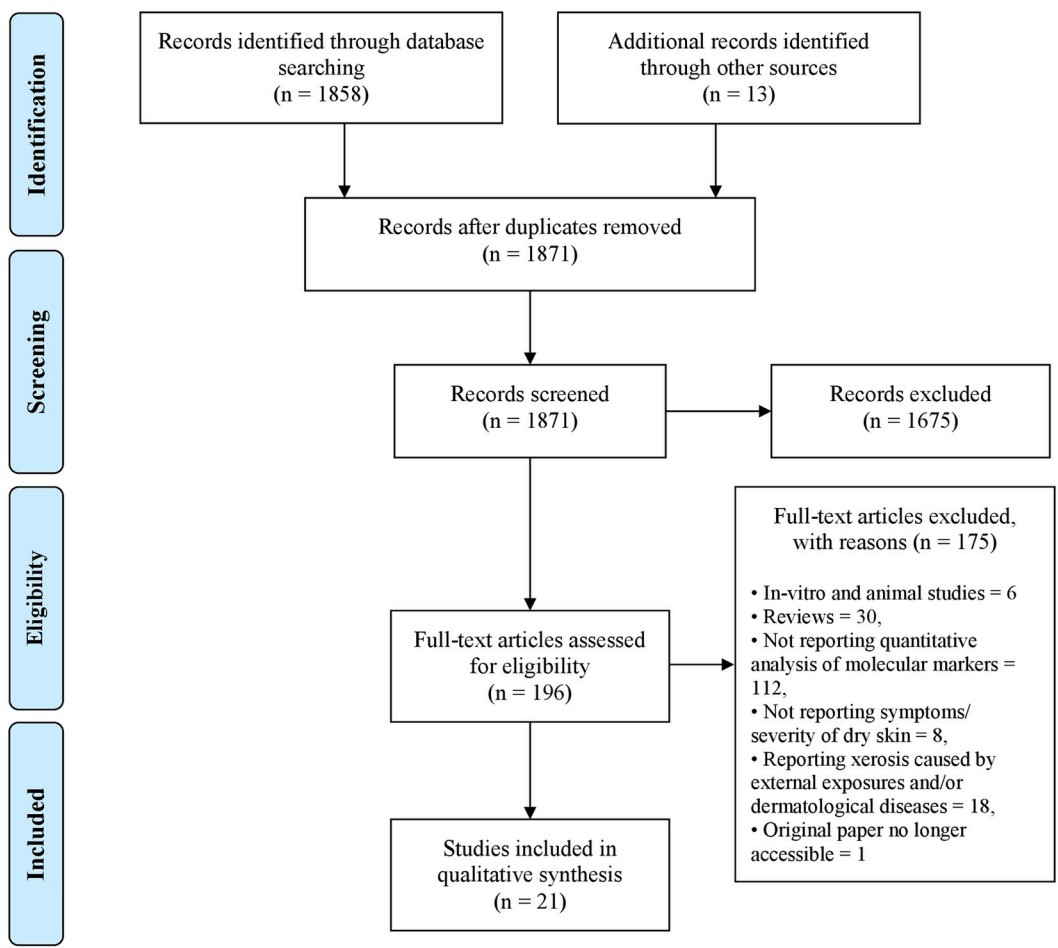

**Fig 1. Flow diagram of the literature search and study selection process.**

One study, which investigated dry skin in cancer patients whose skin dryness was induced by oral intake of erlotinib drug, is reported as drug-induced xerosis [43]. Two studies analyzed markers in the dry skin of patients undergoing hemodialysis [23, 29]. In all other articles, where studies were conducted on apparently healthy participants (not mentioning any underlying internal condition), the subject's skin dryness was referred to as 'general skin dryness'.

### 3.3. Results of individual studies

Study details and results of the data extraction are shown in S2 Appendix. A summary of results is shown in Table 1. Overall, 72 markers were identified. They were sampled from eight skin areas. Most often, liquid chromatography was used as the analytical method. Molecular markers were inductively categorized into (1) lipids, (2) NMFs, (3) proteins and (4) metabolites or metabolic products.

**3.3.1. Lipids.** In different types of dry skin, 25 lipid and lipid like markers were reported. The markers include ceramides (14 parameters), free fatty acids (four parameters), triglyceride, cholesterol, cholesterol sulfate, total lipid, sterol esters, free sterol and wax.

*3.3.1.1. Total ceramide.* All the three studies which analyzed total ceramide in dry skin of patients affected by senile xerosis and diabetic xerosis [24, 28, 41], found this marker to be

**Table 1. Summary of main findings (n = 21 studies).**

| Author | Year | Sample | Age (years) | Skin areas | Method of analysis | Molecular markers analysed | Associations |
|---|---|---|---|---|---|---|---|
| **Hanada et. al.** [23] | 1984 | Haemodialysis patients with dry skin and sweat suppression (n = 5); and healthy volunteers (n = 8) | Not reported | Forearm | Atomic absorption spectrophotometry | Aluminium level in the epidermis | Yes, higher in dry skin of haemodialysis patients |
| | | | | | | Aluminium level in the dermis | |
| **Saint-Léger et. al.** [24] | 1988 | Subjects with xerosis and subjects with normal skin (in total, n = 50) | 25 to 75 | Lateral mid-calf | Photodensitometry | Sterol esters in stratum corneum | Yes, lower in dry skin. |
| | | | | | | Triglycerides in stratum corneum | Yes, lower in dry skin. |
| | | | | | | Polar lipids in stratum corneum | Higher in dry skin. |
| | | | | | | Free fatty acid in stratum corneum | Unclear, higher in dry skin(?) |
| | | | | | | Cholesterol in stratum corneum | No |
| **Horii et. al.** [25] | 1989 | Subjects with mild xerosis (n = 10), moderate xerosis (n = 8), severe xerosis (n = 5) and subjects with normal skin (n = 7) | 59 to 94 | Outer aspect of the lower legs | Amino acid analyser | Amino acid in stratum corneum | Unclear |
| **Saint-Léger et. al.** [26] | 1989 | Subjects with xerosis (n = 52) and subjects with normal skin (n = 12) | 30 to 40 | Outer aspect of the lower legs | Photodensitometry | Wax esters and sterol esters in stratum corneum | Yes, lower in dry skin. |
| | | | | | | Triglycerides in stratum corneum | Unclear, lower in dry skin(?) |
| | | | | | | Free Fatty Acids in stratum corneum | Yes, higher in dry skin. |
| | | | | | | Free sterols in stratum corneum | No |
| | | | | | | Ceramide I in stratum corneum | No |
| | | | | | | Ceramide II in stratum corneum | No |
| | | | | | | Ceramide III in stratum corneum | No |
| | | | | | | Ceramide IV and V in stratum corneum | No |
| | | | | | | Ceramide VI in stratum corneum | No |
| | | | | | | Cholesteryl sulfate in stratum corneum | No |
| | | | | | | Total stratum corneum lipids | No |
| **Jacobson et. al.** [27] | 1990 | Old subjects with dry skin (n = 13) and non-dry skin (n = 7); young subjects with dry skin (n = 8) and non-dry skin (n = 18) | 60 years or older | Outer aspect of the lower legs | High performance liquid chromatography | Aspartic acid in stratum corneum | Unclear, lower in dry skin(?) |
| | | | | | | Threonine in stratum corneum | Unclear |
| | | | | | | Serine in stratum corneum | No |
| | | | | | | Glutamic acid in stratum corneum | Unclear, lower in dry skin(?) |
| | | | | | | Glycine in stratum corneum | Unclear, higher in dry skin(?) |
| | | | | | | Alanine in stratum corneum | Unclear, lower in dry skin(?) |
| | | | | | | Valine in stratum corneum | No |
| | | | | | | Methionine in stratum corneum | No |
| | | | | | | Isoleucine in stratum corneum | Unclear, higher in dry skin(?) |
| | | | | | | Leucine in stratum corneum | Unclear, higher in dry skin(?) |
| | | | | | | Tyrosine in stratum corneum | Unclear, higher in dry skin(?) |
| | | | | | | Phenylalanine in stratum corneum | No |
| | | | | | | Lysine in stratum corneum | No |
| | | | | | | Histidine in stratum corneum | Unclear, lower in dry skin(?) |
| | | | | | | Tryptophan in stratum corneum | No |
| | | | | | | Arginine in stratum corneum | No |
| | | | | | | Ornithine in stratum corneum | No |

(Continued)

**Table 1.** (Continued)

| Author | Year | Sample | Age (years) | Skin areas | Method of analysis | Molecular markers analysed | Associations |
|--------|------|--------|-------------|------------|--------------------|---------------------------|--------------|
| **Akimoto et. al.** [28] | 1993 | Older subjects with xerosis (n = 25), their age matched control (n = 20) and young control group (n = 29) | 24.3 to 71 | Outer aspect of the lower legs | Thin layer chromatography | Total lipid in stratum corneum | Yes, higher in dry skin |
| | | | | | | Total ceramide in stratum corneum | Yes, higher in dry skin |
| | | | | | | Ceramide I in stratum corneum | Yes, higher in dry skin |
| | | | | | | Ceramide II in stratum corneum | Yes, higher in dry skin |
| | | | | | | Ceramide III in stratum corneum | Yes, higher in dry skin |
| | | | | | | Hydro- ceramide I in stratum corneum | Yes, higher in dry skin |
| | | | | | | Ceramide IV and V in stratum corneum | Yes, higher in dry skin |
| | | | | | | Ceramide VI in stratum corneum | Yes, higher in dry skin |
| | | | | | | Cholesterol sulfate | Yes, higher in dry skin |
| | | | | | | Cholesterol Ester | Unclear, lower in dry skin(?) |
| | | | | | | Wax in stratum corneum | Yes, lower in dry skin |
| | | | | | | Triglyceride in stratum corneum | Yes, higher in dry skin |
| | | | | | | Free fatty acid in stratum corneum | Yes, lower in dry skin |
| | | | | | | Cholesterol in stratum corneum | Yes, higher in dry skin |
| **Park et. al.** [29] | 1995 | Patients with xerotic skin undergoing maintenance haemodialysis (n = 10) and healthy volunteers (n = 18) | 30 to 68 | Ventral forearm | Spectrophotometry | Urea in stratum corneum | Yes, higher in dry skin of haemodialysis patients. |
| **Rawlings et. al.** [30] | 1996 | Subjects with dry skin (n = 24) | 23 to 45 | Ventral forearm | Densitometric analysis | Cholesterol in stratum corneum | Yes, lower in dry skin |
| | | | | | Gas chromatography | Fatty acid levels in stratum corneum | Yes, lower in dry skin |
| | | | | | | Ceramide levels in stratum corneum | Yes, lower in dry skin |
| **Schreiner et. al.** [31] | 2000 | Aged subject with dry skin (n = 4), young with dry skin (n = 5) and young with normal skin (n = 10) | 25.5 (SD 2.5) to 66 (SD 3) | Lower leg | High performance thin layer chromatography and Photodensitometry | Total Ceramide in stratum corneum | No |
| | | | | | | Free Sterols in stratum corneum | Yes, lower in dry skin |
| | | | | | | Free fatty acids in stratum corneum | Unclear, higher in dry skin(?) |
| | | | | | | Ceramide (EOS) in stratum corneum | Unclear, higher in dry skin(?) |
| | | | | | | Ceramide (NS) in stratum corneum | Yes, higher in dry skin |
| | | | | | | Ceramide (NP) in stratum corneum | Yes, lower in dry skin |
| | | | | | | Ceramide (EOH) in stratum corneum | Yes, lower in dry skin |
| | | | | | | Ceramide (AS) in stratum corneum | Unclear, higher in dry skin(?) |
| | | | | | | Ceramide (AP) in stratum corneum | Yes, lower in dry skin |
| | | | | | | Ceramide (AH) in stratum corneum | No |
| **Simon et. al.** [32] | 2001 | Xerotic skin (n = 30) and normal skin (n = 26) | 22 to 49 | Outer aspect of the legs | SDS-PAGE, western blotting | Desmoglein 1 in stratum corneum | Yes, higher in dry skin |
| | | | | | | Plakoglobin in stratum corneum | Yes, higher in dry skin |
| | | | | | | Corneodesmosin in stratum corneum | Yes, higher in dry skin |
| | | | | | Transmission electron microscopy | Corneodesmosome density in the inner stratum corneum | Yes, higher in dry skin |
| | | | | | | Corneodesmosome density in the outer stratum corneum | Yes, higher in dry skin |

(Continued)

Table 1. (Continued)

| Author | Year | Sample | Age (years) | Skin areas | Method of analysis | Molecular markers analysed | Associations |
|---|---|---|---|---|---|---|---|
| Takahashi et. al. [33] | 2004 | Aged senile xerosis (n = 12), aged normal (n = 5) and young normal group (n = 10) | 18 to 81 | Lower leg | High performance liquid chromatography | Total amino acid in stratum corneum | Yes, higher in dry skin |
| | | | | | | Aspartic acid in stratum corneum | Unclear, higher in dry skin(?) |
| | | | | | | Glutamic acid in stratum corneum | Unclear, higher in dry skin(?) |
| | | | | | | Citrulline in stratum corneum | Yes, higher in dry skin |
| | | | | | | Serine in stratum corneum | Yes, higher in dry skin |
| | | | | | | Threonine in stratum corneum | Yes, higher in dry skin |
| | | | | | | Arginine in stratum corneum | Yes, lower in dry skin |
| | | | | | | Glycine in stratum corneum | Yes, lower in dry skin |
| | | | | | | Alanine in stratum corneum | Yes, higher in dry skin |
| | | | | | | Proline in stratum corneum | Yes, higher in dry skin |
| | | | | | | Valine in stratum corneum | Yes, higher in dry skin |
| | | | | | | Isoleucine in stratum corneum | Yes, higher in dry skin |
| | | | | | | Leucine in stratum corneum | Yes, higher in dry skin |
| | | | | | | Tryptophan in stratum corneum | Unclear, lower in dry skin(?) |
| | | | | | | Phenylalanine in stratum corneum | Yes, higher in dry skin |
| | | | | | | Urocanic acid in stratum corneum | Yes, higher in dry skin |
| | | | | | | Ornithin in stratum corneum | Yes, higher in dry skin |
| | | | | | | Lysine in stratum corneum | Yes, higher in dry skin |
| | | | | | | Histidine in stratum corneum | Unclear, lower in dry skin(?) |
| | | | | | | Tyrosine in stratum corneum | Unclear, higher in dry skin(?) |
| Delattre et. al. [34] | 2012 | Postmenopausal women and young women with dry skin (n = 10); postmenopausal women and young women with normal skin (n = 10) | 30 to 60 | Upper leg skin | Electrophoresis, western blot, Liquid chromatography mass spectrometry | Corneodesmosin in stratum corneum | Yes, higher in dry skin |
| | | | | | | Annexin A2 in stratum corneum | Yes, higher in dry skin |
| | | | | | | Phosphatidylethanolamine-binding protein 1 (PEBP1) in stratum corneum | Yes, higher in dry skin |
| Ishikawa et. al. [35] | 2013 | Patients with dry skin (n = 20) | 32 to 57 | Outer aspect of the legs | Liquid Chromatography mass spectrometry | Ceramide (NP) in stratum corneum | Yes, lower in dry skin |
| Schweiger et. al. [36] | 2013 | Volunteers with dry and itchy scalp skin (n = 30) | 26 to 73 | Sides of the scalp | Direct analysis in real-Time mass spectrometry | Urea in stratum corneum | Yes, lower in dry scalp skin |
| | | | | | | Lactate in stratum corneum | Yes, lower in dry scalp skin |
| | | | | | Fourier-transformed middle-infrared spectroscopy | Amide band ratio I/II in scalp site | Unclear, lower in dry scalp skin(?) |
| | | | | | | Triglyceride in scalp site | Yes, lower in dry scalp skin |
| | | | | | | Free fatty acid in scalp site | Yes, higher in dry scalp skin |
| | | | | | | Total lipid in stratum corneum | Unclear, lower in dry scalp skin(?) |
| | | | | | Enzyme-linked immunosorbent assays | IL-1ra/IL-1β in stratum corneum | Yes, higher in dry scalp skin |
| | | | | | | IL-8 in stratum corneum | Yes, higher in dry scalp skin |

(Continued)

**Table 1.** (Continued)

| Author | Year | Sample | Age (years) | Skin areas | Method of analysis | Molecular markers analysed | Associations |
|---|---|---|---|---|---|---|---|
| Son et. al. [37] | 2015 | Dry skin and hydrated skin (total n = 22) | Men: 33.8 (5.6), Women: 31.3 (4.1) | Ventral forearm | Western blotting and densitometric analyses | (Pro)filaggrin in stratum corneum | No |
| | | | | | | Bleomycin hydrolase in stratum corneum | Yes, lower in dry skin |
| | | | | | High performance liquid chromatography | Total NMFs (as free amino acid) in stratum corneum | Yes, lower in dry skin |
| | | | | | | Histidine in stratum corneum | Yes, lower in dry skin |
| | | | | | | Serine in stratum corneum | Yes, lower in dry skin |
| | | | | | | Arginine in stratum corneum | Yes, lower in dry skin |
| | | | | | | Glycine in stratum corneum | Yes, lower in dry skin |
| | | | | | | Aspartic acid in stratum corneum | Unclear, lower in dry skin(?) |
| | | | | | | Glutamic acid in stratum corneum | Yes, lower in dry skin |
| | | | | | | Threonine in stratum corneum | Yes, lower in dry skin |
| | | | | | | Alanine in stratum corneum | Yes, lower in dry skin |
| | | | | | | gamma-Aminobutyric acid in stratum corneum | Unclear, lower in dry skin(?) |
| | | | | | | Proline in stratum corneum | Unclear, lower in dry skin(?) |
| | | | | | | Lysine in stratum corneum | No |
| | | | | | | Tyrosine in stratum corneum | Yes, lower in dry skin |
| | | | | | | Methionine in stratum corneum | Yes, lower in dry skin |
| | | | | | | Valine in stratum corneum | Unclear, lower in dry skin(?) |
| | | | | | | Leucine in stratum corneum | Yes, lower in dry skin |
| | | | | | | Isoleucine in stratum corneum | Unclear, lower in dry skin(?) |
| | | | | | | Phenylalanine in stratum corneum | Yes, lower in dry skin |
| | | | | | | Tryptophan in stratum corneum | Yes, lower in dry skin |
| | | | | | | Pyrrolidone carboxylic acid in stratum corneum | Yes, lower in dry skin |
| | | | | | | Urocanic acid in stratum corneum | Unclear, lower in dry skin(?) |
| Danby et. al. [38] | 2016 | Volunteers with dry skin (n = 21) | 60 to 89 | Ventral forearm | Protease assay | Caseinolytic activities in stratum corneum | Yes, higher in dry skin |
| | | | | | | Chymotrypsin-like activities in stratum corneum | Yes, higher in dry skin |
| | | | | | | Trypsin-like activities in stratum corneum | Yes, higher in dry skin |
| | | | | | Fluorometric l -lactate assay | Lactate in stratum corneum | Yes, lower in dry skin |
| | | | | | Not found | Pyrrolidone carboxylic acid (PCA) in stratum corneum | Yes, lower in dry skin |
| | | | | | Fourier transform infrared spectroscopy | Carboxylic acid in stratum corneum | Yes, lower in dry skin |
| Tamura et. al. [39] | 2016 | Subjects having heavily desquamated lips, slightly desquamated lips and subjects having no desquamation on lips (total n = 40) | 22 to 52 | Lips | Liquid chromatography mass spectrometry | Ceramide (NH) in stratum corneum | Yes, lower in dry lip skin |
| | | | | | | Ceramide (NP) in stratum corneum | Yes, lower in dry lip skin |
| Vyumvuhore et. al. [40] | 2018 | Subjects with mild xerosis (n = 19) and subjects with normal skin (n = 15) | 57 and 58 (mean) | On outside arms or the calf | Liquid Chromatography Mass Spectrometry | Ceramide (NdS) in stratum corneum | Yes, lower in dry skin |
| | | | | | | Ceramide (NS) in stratum corneum | Yes, lower in dry skin |
| | | | | | | Ceramide (EOP) in stratum corneum in stratum corneum | Yes, lower in dry skin |

(Continued)

Table 1. (Continued)

| Author | Year | Sample | Age (years) | Skin areas | Method of analysis | Molecular markers analysed | Associations |
|---|---|---|---|---|---|---|---|
| Lechner et. al. [41] | 2019 | Diabetic subjects with xerosis (n = 30) and non-diabetic subjects with xerosis (n = 15) | 63.5 (7.8) and 56.2 (9.3) | Foot dorsum and Plantar heel | Liquid chromatography mass spectrometry | Ceramides | Yes, higher in dry skin of diabetics |
| | | | | | | Natural Moisturising Factors in stratum corneum | Yes, higher in dry skin of diabetics |
| | | | | | | Amino Acid in stratum corneum | Yes, higher in dry skin of diabetics |
| | | | | | | Serine in stratum corneum | Yes, higher in dry skin of diabetics |
| | | | | | | Pyrrolidone carboxylic acid in stratum corneum | Yes, higher in dry skin of diabetics |
| | | | | | | Urocanic acid trans in stratum corneum | Yes, higher in dry skin of diabetics |
| | | | | | | Urocanic acid cis in stratum corneum | No |
| | | | | | | Histamine in stratum corneum | Yes, higher in dry skin of diabetics |
| | | | | | | Total proteins in stratum corneum | Yes, higher in dry skin of diabetics |
| | | | | | | Glutathione in stratum corneum | Unclear |
| | | | | | | Melondialdehyde in stratum corneum | Yes, lower in dry skin of diabetics |
| Legiawati et. al. [42] | 2020 | Type 2 diabetes mellitus patients with dry Skin (total n = 159) | 26 to 59 | Right lower extremities | Enzyme-linked immunosorbent assays | N(6)-carboxymethyl-lysine (CML) activity in stratum corneum | Yes, lower in dry skin of diabetics |
| | | | | | | Interleukin-1α (IL-1α) activity in stratum corneum | Unclear |
| | | | | | | Superoxide dismutase (SOD) activity in stratum corneum | Unclear, lower in dry skin of diabetics(?) |

(Continued)

**Table 1.** (Continued)

| Author | Year | Sample | Age (years) | Skin areas | Method of analysis | Molecular markers analysed | Associations |
|---|---|---|---|---|---|---|---|
| **Uchino et. al.** [43] | 2020 | Patients with non-small lung cancer receiving oral Erlotinib administration having dry skin (n = 18) and healthy subjects (n = 6) | 50 to 85 | Ventral forearm | Ultra performance liquid chromatography combined with time-of-flight mass spectrometry | Cholesterol Sulfate in stratum corneum | Yes, higher in dry skin of patients receiving oral Erlotinib |
| | | | | | | Total free fatty acids in stratum corneum | Unclear, lower in dry skin of patients receiving oral Erlotinib(?) |
| | | | | | | Saturated free fatty acids in stratum corneum | Unclear, lower in dry skin of patients receiving oral Erlotinib(?) |
| | | | | | | Hydroxy free fatty acids in stratum corneum | Unclear, lower in dry skin of patients receiving oral Erlotinib(?) |
| | | | | | | Unsaturated free fatty acids in stratum corneum | No |
| | | | | | | Total Ceramide in stratum corneum | Unclear |
| | | | | | | Ceramide (NdS) in stratum corneum | Unclear, lower in dry skin of patients receiving oral Erlotinib(?) |
| | | | | | | Ceramide (NS) in stratum corneum | Unclear, lower in dry skin of patients receiving oral Erlotinib(?) |
| | | | | | | Ceramide (NP) in stratum corneum | Unclear, lower in dry skin of patients receiving oral Erlotinib(?) |
| | | | | | | Ceramide (NH) in stratum corneum | Unclear, lower in dry skin of patients receiving oral Erlotinib(?) |
| | | | | | | Ceramide (AdS) in stratum corneum | No |
| | | | | | | Ceramide (AS) in stratum corneum | Unclear, lower in dry skin of patients receiving oral Erlotinib(?) |
| | | | | | | Ceramide (AP) in stratum corneum | Yes, lower in dry skin of patients receiving oral Erlotinib |
| | | | | | | Ceramide (AH) in stratum corneum | Unclear, lower in dry skin of patients receiving oral Erlotinib(?) |
| | | | | | | Ceramide (EOdS) in stratum corneum | Unclear, lower in dry skin of patients receiving oral Erlotinib(?) |
| | | | | | | Ceramide (EOS) in stratum corneum | Unclear, lower in dry skin of patients receiving oral Erlotinib(?) |
| | | | | | | Ceramide (EOP) in stratum corneum | Unclear, lower in dry skin of patients receiving oral Erlotinib(?) |
| | | | | | | Ceramide (EOH) in stratum corneum | Unclear, lower in dry skin of patients receiving oral Erlotinib(?) |

higher in those subjects. However, in drug-induced xerosis, association of total ceramide with skin dryness was unclear [43]. In general skin dryness, one study found lower level of total ceramide in the dry skin [30]. Another cross sectional study, conducted in smaller sample size (n = 5 and 10), found no association [31].

*3.3.1.2. Ceramide (NP).* Ceramide (NP), previously known as ceramide III, was found to be lower in three studies regarding general skin dryness [31, 35, 39]. In contrast, one study in older subjects found ceramide (NP) to be remained in higher amount in senile xerosis [28]. Saint léger et. al., 1989 did not found any association of this marker with general skin dryness [26]. In drug-induced xerosis, the association was unclear [43].

*3.3.1.3. Ceramide (NS).* In subjects with senile xerosis, the amount of ceramide (NS), previously ceramide II, was found in lower amounts than their age matched control [28]. Two studies on general skin dryness also found this marker to be associated with dry skin but they reported opposite results to each other [31, 40]. Another study with similar setting did not find any association [26], while in the case of drug-induced xerosis, an association was unclear [43].

*3.3.1.4. Ceramide (EOS), ceramide (NH) and ceramide (EOH).* These three members of ceramide subclasses were found to be positively associated with senile xerosis [28] but negatively associated with general skin dryness [31, 39, 40]. However, one study showed no association of these ceramides with general skin dryness [26] and another study showed it to be unclear [43].

*3.3.1.5. Ceramide (AS) and hydroceramide I.* Ceramide (AS) and hydroceramide I were only found to be associated with senile xerosis and the reported amount was higher in the aged dry skin [28]. However, additional studies which analyzed ceramide (AS) in other dry skin conditions (general skin dryness and drug-induced xerosis), reported either unclear or no association [26, 31, 43].

*3.3.1.6. Ceramide (AP) and ceramide (NdS).* All the studies that analyzed the quantitative amounts of these two ceramides, reported these markers to be present in lower amounts in different dry skin conditions. Ceramide (AP) was investigated both in general skin dryness and drug-induced xerosis [31, 43] while ceramide (NdS) was only analyzed in general skin dryness [40].

*3.3.1.7. Ceramide (AH), ceramide (AdS), ceramide (EOdS) and ceramide (EOP).* No study reported any positive or negative association of these four ceramides with any type of xerosis cutis.

*3.3.1.8. Total free fatty acids.* Seven studies published between 1988 and 2020 analyzed total free fatty acids, of which four reported associations of this marker with different dry skin conditions [26, 28, 30, 36]. Akimoto et. al., 1993 found the amount of free fatty acid to be lower in older subjects with xerosis than their age matched control [28]. Two studies on general skin dryness (one cross sectional, another, randomized controlled trial) found opposite results to each other; higher [26] and lower [30]. The amount of free fatty acids were found higher in dry and itchy scalp skin compared to the side of the scalp which achieved reduced dryness after a tonic treatment [36]. Results reported by other three studies were found to be unclear [24, 31, 43]. Uchino et. al., 2020 [43] also analyzed three categories of free fatty acids in the dry skin of patients receiving erlotinib drug. Unsaturated free fatty acids were not associated with drug-induced xerosis while saturated and hydroxyl free fatty acids revealed unclear association.

*3.3.1.9. Triglycerides.* Two studies on senile xerosis reported the association of triglycerides with skin dryness. One study found this to be higher in aged dry skin compared to the control sample while another study found the opposite [24]. In general skin dryness, one study found no association [26] but in dry scalp skin, the amount of triglycerides was comparatively lower when the scalp was found to be drier [36].

*3.3.1.10. Cholesterol and cholesterol sulfate.* Studies, where an association was present, both of these two markers were shown to be in lower amounts in general skin dryness [30] and in higher amounts in senile xerosis and drug-induced xerosis [28, 43]. However, there is also one

study per marker, which reported no association of cholesterol and the sulfate ester of this compound with dry skin.

*3.3.1.11. Free sterols, sterol esters and wax.* Like cholesterol, total free sterols and total sterol esters were also found to be in lower amounts in general skin dryness [26, 31], but unlike the sulfate ester, total sterol esters [24] and wax [28] were found to be in lower amounts in senile xerosis [24, 28]. There are also other studies in this review, which reported unclear association of sterol esters in senile xerosis [28] and no association of free sterols in senile xerosis [26].

*3.3.1.12. Total lipids.* Three studies reported this marker, one study described an association [28], one described an unclear association [36] and the remaining study described no association [26] with skin dryness. In the study where an association was found, a higher amount of total lipid in senile xerosis was reported [28].

**3.3.2. Natural moisturizing factors (NMFs).** Twenty-five NMFs components were reported in different dry skin etiologies, which include most standard amino acids, ornithin, citrulline, gamma-aminobutyric acid, urocanic acid, carboxylic acids and pyrrolidone carboxylic acid.

*3.3.2.1. Total free amino acids (FAAs) and NMFs.* Total FAA was found to be higher in the dry skin of patients with underlying conditions like senile xerosis [33] and diabetic xerosis [41]. Analysis of NMFs also revealed the same pattern [41]. Inversely, in general skin dryness, the amount of FFAs was found to be lower than the control samples [37]. One study, however, found unclear association of FAAs in senile xerosis [25].

*3.3.2.2. Serine, alanine, leucine, phenylalnine and threonine.* These five amino acids followed the similar pattern as total FAAs. Amounts of these amino acids were higher in senile xerosis and diabetic xerosis [33, 41] and were lower in general skin dryness [37]. However there is at least one study which found either 'unclear' or 'no' association of these amino acids with general skin dryness [27].

*3.3.2.3. Glycine and arginine.* In both senile xerosis and general skin dryness, glycine and arginine was negatively associated [33, 37], hence, amounts were found to be lower than in the control group. Unclear or no association of these two amino acids were also reported [27].

*3.3.2.4. Histidine, tyrosine, glutamic acid, tryptophan and methionine.* For these five amino acids, association was reported only in case of general skin dryness and the amounts were lower compared to the control group [37]. One study on senile xerosis [33] and another study on general skin dryness [27], both worked on small control groups (n = 5 and 7), reported either 'unclear' or 'no' association of these amino acids with xerosis cutis.

*3.3.2.5. Isoleucine, valine, lysine, proline, ornithin and citrulline.* All these six amino acids were reported to be associated with only senile xerosis [33]. The association was positive; that means in aged skin, these amino acids were found to be in higher amounts than the control samples. Except citrulline, other five amino acids were showed to have either 'unclear' or 'no' association with general skin dryness [27, 37].

*3.3.2.6. Aspartic acid and gamma-aminobutyric acid.* Only unclear associations were found in general skin dryness [27, 37] and senile xerosis [33].

*3.3.2.7. Urocanic acid, carboxylic acids and pyrrolidone carboxylic acid (PCA).* Urocanic acid was reported to be present in higher amounts in senile xerosis [33] and also in diabetic xerosis [41]; as trans urocanic acid. However, in case of cis urocanic acid, no association was found with diabetic xerosis [41]. In general skin dryness, the association was not clear [37]. Carboxylic acids (total) followed different pattern- 'negative association' with senile xerosis [38]. When only pyrrolidone carboxylic acid was investigated, it was reported to be present in lower amounts in general skin dryness and senile xerosis [37, 38] but in higher amounts in diabetic xerosis [41].

**3.3.3. Proteins/ enzymes.** Described below are the 17 protein, enzyme, cytokines and similar markers which were reported in the included articles in this review.

*3.3.3.1. Corneodesmosin, desmoglein 1, plakoglobin, annexin A2 and phosphatidylethanolamine-binding protein 1.* These five protein markers were found to be positively associated with general skin dryness. Corneodesmosin was investigated in two studies [32, 34] while the others were studied once [32] or [34]. In all cases, the amount of these proteins where quantified in higher amounts in dry skin compared to the subjects' age-matched control. It is to be noted that in the study by Delattre et. al. 2012, who analyzed corneodesmosin, annexin A2 and phosphatidylethanolamine-binding protein 1, about half of the study population was postmenopausal women [34].

*3.3.3.2. Caseinolytic activities, chymotrypsin-like activities, trypsin-like activities and total proteins.* These four protein markers were found to be in elevated amounts in dry skin of patients with underlying conditions. Caseinolytic activities, chymotrypsin-like activities and trypsin-like activities were measured in senile xerosis [38]. These markers were positively associated with skin dryness. Total protein was shown to be increased in diabetic xerosis [41].

*3.3.3.2. N(6)-carboxymethyl-lysine activity and bleomycin hydrolase.* Being negatively associated with dry skin, N(6)-carboxymethyl-lysine activity was reported in diabetic xerosis [42] and bleomycin hydrolase was reported in general skin dryness [37]. In both cases, amount of these markers were found to be in lower amount in dry skin compared to the control groups.

*3.3.3.3. Glutathione, (pro)filaggrin and superoxide dismutase activity.* Glutathione, a tri-peptide, was detected in non-diabetics with dry skin though it was not found in diabetics with dry skin [41]. The association seems unclear. (Pro)filaggrin was also reported to have no association in general skin dryness [37]. The association of superoxide dismutase was unclear with diabetic xerosis as reported by Legiawati et. al., 2020 [42].

*3.3.3.4. Cytokines (Interleukin (IL)-8, IL-1ra/IL-1β and Interleukin-1α).* In scalp skin (general skin dryness), the amount of interleukin-8 was found to be higher in the dry scalp compared to the amount of this marker found in the hydrated scalp after tonic treatment. The ratio of IL-1ra/IL-1β was also positively associated with scalp dryness [36]. Another study which measured interleukin-1α activity in diabetic xerosis, found its association with the skin dryness to be unclear [42].

**3.3.4. Metabolites or metabolic products.** Five metabolites/ metabolic products including lactate, urea, histamine, melondialdehyde and aluminium were reported to be associated with dry skin.

*3.3.4.1. Lactate.* Both of the two studies which investigated on the amount of lactate in the skin, found this marker to be negatively associated with skin dryness. One study was on dry scalp skin (general skin dryness) [36] and another was on senile xerosis [38].

*3.3.4.2. Urea.* In the dry skin of patients undergoing hemodialysis, the amount of urea was found to be higher compared to control subjects [29]. The opposite was found in case of dry scalp skin (general skin dryness) where the amount of urea was negatively associated with dryness of scalp [36].

*3.3.4.3. Histamine and melondialdehyde.* Both of these markers were shown to be associated with the dry skin of diabetic patients compared to skin dryness in non-diabetics. Histamine, a neurotransmeter, was positively associated with diabetic xerosis while melondialdehyde, a marker of oxidative stress, was decreased in diabetic xerosis [41].

*3.3.4.4. Aluminium.* In the dry skin of hemodialysis patients, aluminium levels in the epidermis and dermis were higher than in the control group and seemed to be positively associated with the skin dryness [23].

### 3.4. Number of markers and possible associations with dry skin

Table 2 presents a summary of all molecular markers, which were reported at least in two studies (top markers). Additionally, S3 Appendix is for the markers which was analyzed only in one study. Total free fatty acids, total ceramide, ceramide (NP), ceramide (NS), ceramide (NH), ceramide (EOS), ceramide (EOH), ceramide (AS), triglyceride, total free amino acids, serine and urocanic acid were measured in at least four studies. From those, the number of studies suggesting associations between molecular markers and dry skin compared to the number of studies of unclear or no associations was higher for total free fatty acids, total ceramide, ceramide (NP), ceramide (NS), triglyceride, total free amino acids and serine.

## 4. Discussion

This systematic review identified more than 70 molecular markers that were measured in dry skin research. In addition, various sampling and analytical methods were used. Overall, only 12 molecular markers were reported in at least four studies. The majority of markers was reported only once or twice. This indicates substantial heterogeneity in this field and makes the intended comparisons nearly impossible.

When considering the markers, which were reported at least four times, seven seemed to be associated with skin dryness in at least two or more studies (total ceramide, ceramide (NP), ceramide (EOS), ceramide (NH), ceramide (EOH), free amino acids and serine). If associated, they were always found to be lower in general skin dryness but higher in xerosis induced by any internal condition. Additional markers, which seem to show a similar pattern are cholesterol, cholesterol sulfate, alanine, leucine, phenylalanine, threonine and urea. Though these were analyzed in less number of studies, associations with xerosis cutis were reported in at least two studies. In addition, the independent association of ceramide (NP), ceramide (NH) and cholesterol sulfate was demonstrated by statistical analysis in corresponding studies [35, 39, 43].

Total free fatty acids, ceramide (NS) and triglycerides were also analyzed in four or more studies but the associations of these markers with xerosis cutis seemed unclear. For example, in general skin dryness, total free fatty acids were shown to have both positive [26, 36] and negative associations [30]. Same was also seen for ceramide (NS) [31, 40]. Triglycerides in senile xerosis also showed conflicting results [24, 28]. Moreover, for nearly every marker there were also studies showing unclear or no association. In addition to the wide variety of reported markers, this may indicate substantial biological variability. Variations may be caused by the analytical methods (e.g., SC or compounds dissolved from SC) used. In addition, use of different sampling methods (tape-stripping, varnish stripping, solvent extraction, etc) might contribute to the variability in results. Sensitivity differences among individual methods of analysis may produce remarkable variability as only six recent studies used unambiguous quantitation technology like mass spectrometry while others used different spectrophotometric techniques such as photodensitometry, thin layer chromatography, liquid chromatography, gas chromatography or other biomolecular tools depending on the analyte characteristics. Moreover, variations in study design, number of samples and reported quantitative units might also have contributed to observed heterogeneity and variability to some extent.

We also found four markers (pyrrolidone carboxylic acid, corneodesmosin, lactate and urea) which were associated with dry skin in all the few studies they were reported. PCA was analyzed in three studies with both negative [37, 38] and positive [41] association. Corneodesmosin was found to be positively associated [32, 34] while lactate [36, 38] and urea [29, 36]

**Table 2. Top markers (compounds analysed more than once).**

| Molecular markers | Number of studies | Analysed material | Sampling technique | Method of analysis | Association with skin dryness (number of studies) |
|---|---|---|---|---|---|
| Total free fatty acids [24, 26, 28, 30, 31, 36, 43] | 7 | Compounds dissolved from stratum corneum/ stratum corneum/ direct measurement of skin area | Hexane- methanol extraction/ stripping with cyanoacrylate resin / tape stripping/ shave biopsy/ direct measurement | Photodensitometry/ thin layer chromatography/ fourier-transformed middle-infrared spectroscopy/ high performance thin layer chromatography/ liquid chromatography mass spectrometry | Yes: 4 Unclear: 3 |
| Total ceramide [24, 28–31, 41, 43] | 6 | Compounds dissolved from stratum corneum/stratum corneum | Hexane- methanol extraction/ stripping with cyanoacrylate resin / tape stripping/ shave biopsy/ collecting swabs. | Photodensitometry/ thin layer chromatography/ high performance thin layer chromatography/ liquid chromatography mass spectrometry | Yes: 4 / No: 1 / Unclear: 1 |
| Ceramide (NP); also called Ceramide III. [19, 26, 28, 31, 35, 39, 43] | 6 | Compounds dissolved from stratum corneum/stratum corneum | Hexane- methanol extraction/ stripping with cyanoacrylate resin / shave biopsy/ varnish stripping/ tape stripping | Photodensitometry/ thin layer chromatography/ high performance thin layer chromatography/ liquid chromatography mass spectrometry | Yes: 4 / No: 1 / Unclear: 1 |
| Ceramide (NS); also called Ceramide II. [19, 26, 28, 31, 40, 43] | 5 | Compounds dissolved from stratum corneum/stratum corneum | Hexane- methanol extraction/ stripping with cyanoacrylate resin / shave biopsy/ collecting swabs/ tape stripping. | Photodensitometry/ thin layer chromatography/ high performance thin layer chromatography/ liquid chromatography mass spectrometry | Yes: 3 / No: 1 / Unclear: 1 |
| Ceramide (EOS); also called Ceramide I. [19, 26, 28, 31, 40, 43] | 5 | Compounds dissolved from stratum corneum/stratum corneum | Hexane- methanol extraction/ stripping with cyanoacrylate resin / shave biopsy/ collecting swabs/ tape stripping. | Photodensitometry/ thin layer chromatography/ high performance thin layer chromatography/ liquid chromatography mass spectrometry | Yes: 2 / No: 1 / Unclear: 2 |
| Triglyceride [24, 26, 28, 36] | 4 | Compounds dissolved from stratum corneum/ stratum corneum/ direct measurement of skin area | Hexane- methanol extraction/ stripping with cyanoacrylate resin / direct measurement | Photodensitometry/ thin layer chromatography/ fourier-transformed middle-infrared spectroscopy | Yes: 3 / Unclear: 1 |
| Serine [27, 33, 37, 41] | 4 | Scraped cells from stratum corneum/ stratum corneum/ compounds dissolved from stratum corneum. | Scraping off the skin with a glass slide/ tape stripping/ collecting swabs | High performance liquid chromatography/ liquid chromatography mass spectrometry | Yes: 3 / No: 1 |
| Total free amino acids [25, 33, 37, 41] | 4 | Stratum corneum/ compounds dissolved from stratum corneum | Tape stripping/ scraping off the skin with a glass slide/ collecting swabs | Amino acid analyzer/ high performance liquid chromatography/ liquid chromatography mass spectrometry | Yes: 3 / Unclear: 1 |
| Ceramide (NH); also called Ceramide VI [19, 26, 28, 39, 43] | 4 | Compounds dissolved from stratum corneum/stratum corneum | Hexane- methanol extraction/ stripping with cyanoacrylate resin/ tape stripping | Photodensitometry/ thin layer chromatography/ liquid chromatography mass spectrometry | Yes: 2 / No: 1 / Unclear: 1 |
| Urocanic acid (UCA) [27, 33, 37, 41] | 4 | Stratum corneum/ compounds dissolved from stratum corneum | Scraping off the skin with a glass slide/ tape stripping/ collecting swabs | High performance liquid chromatography/ liquid chromatography mass spectrometry | Yes: 2 (1 as UCA trans) / No: 1 (as UCA cis) / Unclear: 1 |
| Ceramide (EOH); also called Ceramide IV. [26, 28, 31, 43] | 4 | Compounds dissolved from stratum corneum/stratum corneum | Hexane- methanol extraction/ stripping with cyanoacrylate resin / shave biopsy/ tape stripping. | Photodensitometry/ thin layer chromatography/ high performance thin layer chromatography/ liquid chromatography mass spectrometry | Yes: 1 / No: 1 / Unclear: 2 |
| Ceramide (AS) [26, 28, 31, 43] | 4 | Compounds dissolved from stratum corneum/stratum corneum | Hexane- methanol extraction/ stripping with cyanoacrylate resin / shave biopsy/ tape stripping. | Photodensitometry/ thin layer chromatography/ high performance thin layer chromatography/ liquid chromatography mass spectrometry | No: 1 / Unclear: 3 |
| Pyrrolidone carboxylic acid [37, 38, 41] | 3 | Stratum corneum/ compounds dissolved from stratum corneum | Tape stripping/ collecting swabs | High performance liquid chromatography / liquid chromatography mass spectrometry | Yes: 3 |

(*Continued*)

**Table 2.** (Continued)

| Molecular markers | Number of studies | Analysed material | Sampling technique | Method of analysis | Association with skin dryness (number of studies) |
|---|---|---|---|---|---|
| Glycine [27, 33, 37] | 3 | Stratum corneum | Scraping off the skin with a glass slide/ tape stripping | High performance liquid chromatography | Yes: 2<br>Unclear: 1 |
| Alanine [27, 33, 37] | 3 | Stratum corneum | Scraping off the skin with a glass slide/ tape stripping | High performance liquid chromatography | Yes: 2<br>Unclear: 1 |
| Leucine [27, 33, 37] | 3 | Stratum corneum | Scraping off the skin with a glass slide/ tape stripping | High performance liquid chromatography | Yes: 2<br>Unclear: 1 |
| Phenylalaine [27, 33, 37] | 3 | Stratum corneum | Scraping off the skin with a glass slide/ tape stripping | High performance liquid chromatography | Yes: 2<br>No: 1 |
| Arginine [27, 33, 37] | 3 | Stratum corneum | Scraping off the skin with a glass slide/ tape stripping | High performance liquid chromatography | Yes: 2<br>No: 1 |
| Threonine [27, 33, 37] | 3 | Stratum corneum | Scraping off the skin with a glass slide/ tape stripping | High performance liquid chromatography | Yes: 2<br>Unclear: 1 |
| Cholesterol [24, 28, 30] | 3 | Compounds dissolved from stratum corneum/stratum corneum | Hexane- methanol extraction/ stripping with cyanoacrylate resin / tape stripping | Photodensitometry/ thin layer chromatography | Yes: 2<br>No: 1 |
| Cholesterol sulfate [26, 28, 43] | 3 | Compounds dissolved from stratum corneum/stratum corneum | Hexane- methanol extraction/ stripping with cyanoacrylate resin / tape stripping | Photodensitometry/ thin layer chromatography/ liquid chromatography mass spectrometry | Yes: 2<br>No: 1 |
| Corneodesmosin [32, 34] | 2 | Stratum corneum | Varnish stripping | Electrophoresis, western blot and liquid chromatography mass spectrometry. | Yes: 2 |
| Lactate [36, 38] | 2 | Compounds dissolved from stratum corneum | Skin surface material collected by DIP-it sampler/ collecting swabs | Real-time mass spectrometry/ fluorometric L -lactate assay. | Yes: 2 |
| Urea [29, 36] | 2 | Stratum corneum/ compounds dissolved from stratum corneum | Cyanoacrylate adhesive stripping/ skin surface material collected by DIP-it sampler | Spectrophotometry/ real-time mass spectrometry | Yes: 2 |
| Ceramide (AP) [31, 43] | 2 | Stratum corneum | Shave biopsy/ tape stripping. | High performance thin layer chromatography and photodensitometry/ liquid chromatography mass spectrometry | Yes: 2 |
| Histidine [27, 33, 37] | 3 | Stratum corneum | Scraping off the skin with a glass slide/ tape stripping | High performance liquid chromatography | Yes: 1<br>Unclear: 2 |
| Tyrosine [27, 33, 37] | 3 | Stratum corneum | Scraping off the skin with a glass slide/ tape stripping | High performance liquid chromatography | Yes: 1<br>Unclear: 2 |
| Glutamic acid [27, 33, 37] | 3 | Stratum corneum | Scraping off the skin with a glass slide/ tape stripping | High performance liquid chromatography | Yes: 1<br>Unclear: 2 |
| Isoleucine [27, 33, 37] | 3 | Stratum corneum | Scraping off the skin with a glass slide/ tape stripping | High performance liquid chromatography | Yes: 1<br>Unclear: 2 |
| Tryptophan [27, 33, 37] | 3 | Stratum corneum | Scraping off the skin with a glass slide/ tape stripping | High performance liquid chromatography | Yes: 1<br>No: 1<br>Unclear: 1 |
| Valine [27, 33, 37] | 3 | Stratum corneum | Scraping off the skin with a glass slide/ tape stripping | High performance liquid chromatography | Yes: 1<br>No: 1<br>Unclear: 1 |
| Total lipid [26, 28, 36] | 3 | Compounds dissolved from stratum corneum/ stratum corneum/ direct measurement of skin area | Hexane- methanol extraction/ stripping with cyanoacrylate resin / direct measurement | Photodensitometry/ thin layer chromatography/ fourier-transformed middle-infrared spectroscopy | Yes: 1<br>No: 1<br>Unclear: 1 |

(*Continued*)

**Table 2.** (Continued)

| Molecular markers | Number of studies | Analysed material | Sampling technique | Method of analysis | Association with skin dryness (number of studies) |
|---|---|---|---|---|---|
| Lysine [27, 33, 37] | 3 | Stratum corneum | Scraping off the skin with a glass slide/ tape stripping | High performance liquid chromatography | Yes: 1<br>Unclear: 2 |
| Sterol esters [24, 26, 28] | 3 | Compounds dissolved from stratum corneum/ stratum corneum | Hexane- methanol extraction/ stripping with cyanoacrylate resin | Photodensitometry/ thin layer chromatography | Yes: 1<br>Unclear: 2 |
| Proline [33, 37] | 2 | Stratum corneum | Scraping off the skin with a glass slide/ tape stripping | High performance liquid chromatography | Yes: 1<br>Unclear: 1 |
| Ceramide (NdS) [40, 43] | 2 | Compounds dissolved from stratum corneum/stratum corneum | Collecting swabs/ tape stripping. | Liquid chromatography mass spectrometry | Yes: 1<br>Unclear: 1 |
| Methionine [27, 37] | 2 | Stratum corneum | Scraping off the skin with a glass slide/ tape stripping | High performance liquid chromatography | Yes: 1<br>No: 1 |
| Ornithin [27, 33] | 2 | Stratum corneum | Scraping off the skin with a glass slide | High performance liquid chromatography | Yes: 1<br>No: 1 |
| Free sterols [26, 31] | 2 | Compounds dissolved from stratum corneum/stratum corneum | Hexane- methanol extraction/ shave biopsy | Photodensitometry/high performance thin layer chromatography | Yes: 1<br>No: 1 |
| Aspartic acid [27, 33, 37] | 3 | Stratum corneum | Scraping off the skin with a glass slide/ tape stripping | High performance liquid chromatography | Unclear: 3 |
| Ceramide (AH) [31, 43] | 2 | Stratum corneum | Shave biopsy/ tape stripping. | High performance thin layer chromatography and photodensitometry/ liquid chromatography mass spectrometry | No: 1<br>Unclear: 1 |

were found to be negatively associated with skin dryness. More studies are required to evaluate the significance of these markers.

Quantitative expressions of several markers were found to be consistently changing with multiple clinical score values of skin dryness in corresponding samples. Triglycerides, ceramide (NH), ceramide (NP), ceramide (AP), urea and lactate showed gradual increase; while total free fatty acids and cholesterol sulfate were found to be gradually decreased with the reported severities of dry skin assessed according to the scoring methods. However, except urea and lactate (though reported in only two studies), other studies reported unclear or no associations of these markers which indicates heterogeneity in overall expression.

In case of dry skin induced by internal diseases, markers of diabetic xerosis was studied exhaustively in two recent studies by Lechner et. al., 2019 [41] and Legiawati et. al., 2020 [42]. Among the markers, pyrrolidone carboxylic acid was higher in diabetic xerosis; but in other dry skin conditions (general skin dryness and senile xerosis), there were negative associations. Trans-urocanic acid was positively associated but cis-urocanic acid was not associated with diabetic xerosis. Total ceramide, NMFs and histamine were positively associated while N(6)-carboxymethyl-lysine and melondialdehyde was negatively associated.

It is also well known, that the occurrence and severity of xerosis cutis is skin area specific, for example in senile xerosis the legs are drier than the arms [2]. However, the heterogeneity of the reviewed evidences makes these intended comparisons almost impossible. In addition, we did not include any study that compared skin dryness or markers from both the arms and leg skin areas.

Further research in this field is necessary to facilitate the discovery of evidence of associations of the molecular markers with skin dryness and to help in guiding clinical practice. The status of certain markers may even help clinicians in more precise understanding of the underlying causes of the disease. However, for translating the research findings into clinical practice, as recommended by Hammond and Taube [44], the markers should be validated in prospective, well-controlled clinical trials of various patient participants across different institutions with established standard for sample preparations, data collection, statistical analysis and scoring. Many studies analyzed multiple markers simultaneously. Besides considering the individual markers, a panel of markers might also provide a better inside in disease prognosis especially in xerosis cutis with underlying conditions, which merits further investigation.

One of the limitations of this systematic review is that we selected the top markers primarily based on the number of articles in which they were analyzed. We searched for particular patterns regarding the occurrence of the markers with the presence or severity of skin dryness. That is why the markers, which were analyzed only in one study, could not be placed as top markers though some might have potential as important markers. The objective of this review was to describe possible associations of molecular markers based on their quantitative patterns related to skin dryness. To define the association, an arbitrary evaluation of the patterns was used which is another limitation of this study. In addition, as the p-values are affected by the sample size, we considered the difference between the quantitative amounts of the markers found in the comparing groups rather than the reported p-values which were actually present only in few articles and unlikely to be clinically relevant. Additional limitation of this study is that, group comparisons between the skin of healthy people and the skin of people with underlying conditions might be biased as they also differ in other characteristics beyond skin dryness (diabetes, hemodialysis, hormonal imbalance, drug effects, etc.). Also, we did not include temporary skin dryness due to seasonal changes which is more logical to be described as rough skin as stated by De Paepe et.al., 2009 [45]. As we were interested in reviewing the markers studied in pathological xerosis, seasonal dry skin was not in our focus.

## 5. Conclusion

Seventy-two molecular markers for measuring xerosis cutis were identified. Total free fatty acids, ceramides, triglycerides, total free amino acids, serine and urocanic acid have been reported most often, but the evidence whether the quantity of these molecular markers indicates the status of skin dryness is heterogeneous. Thirty-one molecular markers were reported only once. Although there is a huge interest in molecular markers in dry skin research, it is currently unclear which are the most relevant.

## Supporting information

**S1 Appendix. Search strategy.**
(PDF)

**S2 Appendix. Study details and results of the data extraction.**
(DOCX)

**S3 Appendix. Molecular markers analyzed only once.**
(DOCX)

**S4 Appendix. PRISMA checklist.** A review protocol has been registered in the PROSPERO database (https://www.crd.york.ac.uk/prospero/display_record.php?ID=CRD42020214173). (DOCX)

**S1 File.**
(PDF)

## Author Contributions

**Conceptualization:** Ruhul Amin, Jan Kottner.

**Data curation:** Ruhul Amin, Anna Lechner.

**Formal analysis:** Ruhul Amin, Anna Lechner, Jan Kottner.

**Methodology:** Anna Lechner, Jan Kottner.

**Resources:** Ulrike Blume-Peytavi, Jan Kottner.

**Software:** Ulrike Blume-Peytavi, Jan Kottner.

**Supervision:** Annika Vogt, Ulrike Blume-Peytavi, Jan Kottner.

**Writing – original draft:** Ruhul Amin.

**Writing – review & editing:** Anna Lechner, Annika Vogt, Ulrike Blume-Peytavi, Jan Kottner.

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
