## [Decision Letter · Decision Letter 0]

13 Oct 2021

PONE-D-21-19523Molecular characterization of xerosis cutis: a systematic reviewPLOS ONE

Dear Dr. Amin,

Thank you for submitting your manuscript to PLOS ONE. After careful consideration, we feel that it has merit but does not fully meet PLOS ONE’s publication criteria as it currently stands. Therefore, we invite you to submit a revised version of the manuscript that addresses all the minor points raised during the review process.

Please submit your revised manuscript by Nov 27 2021 11:59PM.  Please include the following items when submitting your revised manuscript:A rebuttal letter that responds to each point raised by the academic editor and reviewer(s). You should upload this letter as a separate file labeled 'Response to Reviewers'.A marked-up copy of your manuscript that highlights changes made to the original version. You should upload this as a separate file labeled 'Revised Manuscript with Track Changes'.An unmarked version of your revised paper without tracked changes. You should upload this as a separate file labeled 'Manuscript'.

We look forward to receiving your revised manuscript.

Kind regards,

Michel Simon, Ph. D.

Academic Editor

PLOS ONE

Journal Requirements:

This study was supported by Charité-Universitätsmedizin Berlin, Department of Dermatology, Venereology and Allergology, Clinical Research Center for Hair and Skin Science, Germany. RA receives scholarship from ‘Bangabandhu Science and Technology Fellowship trust, Ministry of Science and Technology, Bangladesh’ to conduct his doctoral study in Charité-Universitätsmedizin Berlin. The funders had no role in study design, data collection and analysis, decision to publish, or preparation of this manuscript.

This study was supported by Charité-Universitätsmedizin Berlin, Department of Dermatology, Venereology and Allergology, Clinical Research Center for Hair and Skin Science, Germany. RA receives scholarship from ‘Bangabandhu Science and Technology Fellowship trust, Ministry of Science and Technology, Bangladesh’ to conduct his doctoral study in Charité-Universitätsmedizin Berlin. The funders had no role in study design, data collection and analysis, decision to publish, or preparation of this manuscript.

Additional Editor Comments:

This review about xerosis cutis biomarkers is of potential value for cosmetic but also pharmaceutic and basic research. The review seems to be comprehensive. I have few minor points, as follows.

- Please use systematically the more accurate term to define the particular form of xerosis cutis you are speaking about (senile xerosis, drug induced xerosis, general skin dryness, etc.); without this precision in mind, miss-interpretations of the data could occur. For example, line 214, § “Ceramide-EOS, -NH and –EOH, the term xerosis cutis is not enough precise. May be a paragraph specifically dedicated to the description fo xerosis cutis forms would be informative.

- It would probably be better to use the term “carboxylic acids” (plural) when a particular carboxylic acid is not looked for, e.g., line 309.

- line 286, please change “the is” to ‘there is”.

Reviewers' comments:

Reviewer's Responses to Questions

**Comments to the Author**

1. Is the manuscript technically sound, and do the data support the conclusions?

Reviewer #1: Yes

2. Has the statistical analysis been performed appropriately and rigorously? 

Reviewer #1: N/A

3. Have the authors made all data underlying the findings in their manuscript fully available?

Reviewer #1: No

4. Is the manuscript presented in an intelligible fashion and written in standard English?

Reviewer #1: Yes

5. Review Comments to the Author

Reviewer #1: This is an interesting systematic review. I think it would be useful to include the keywords used in the search in the main paper as this is key. In general legs are drier than forearms. It might be interesting to comment in the discussion on any differences observed between studies where leg skin was used versus arm skin. For completion, the PROSPERO details should be included in the supplementary data.

6. PLOS authors have the option to publish the peer review history of their article (what does this mean?). If published, this will include your full peer review and any attached files.

Reviewer #1: No

---

## [Author Response · Author response to Decision Letter 0]

20 Oct 2021

Response to reviewers

Dear academic editor and reviewer,

Thank you very much for assessing our manuscript and providing your recommendations. Please find our point-by-point responses below. Changes in the manuscript text are highlighted in yellow.

Journal Requirements:

Comment #1: Please ensure that your manuscript meets PLOS ONE's style requirements, including those for file naming.

Response: We rechecked the manuscript documents according to the PLOS ONE style templates. Street address with postal code was deleted.

Changes to manuscript: 1Charité-Universitätsmedizin Berlin, corporate member of Freie Universität Berlin and Humboldt-Universität zu Berlin, Department of Dermatology, Venereology and Allergology, Clinical Research Center for Hair and Skin Science, Berlin, Germany. (Author affiliations, page 1, line 6)

Comment #2: Please note that funding information should not appear in the Acknowledgments section or other areas of your manuscript. … Please remove any funding-related text from the manuscript and let us know how you would like to update your Funding Statement.

Response: We have removed any formal acknowledgment section in our manuscript. Funding Statement for the online submission should be shortened without changing the originally provided information. Please consider publishing the funding statement as following: "This study was conducted in Charité-Universitätsmedizin Berlin, Department of Dermatology, Venereology and Allergology, Clinical Research Center for Hair and Skin Science, Germany. RA acknowledges the doctoral scholarship support of Bangabandhu Science and Technology Fellowship trust, Ministry of Science and Technology, Bangladesh".

Changes to manuscript: The paragraph under the heading ‘funding’ was deleted.

Comment #3: Please note that in order to use the direct billing option the corresponding author must be affiliated with the chosen institute.

Response: The corresponding author is affiliated with the chosen institute.

Changes to manuscript: None.

Comment #4: Please review your reference list to ensure that it is complete and correct.

Response: The reference list was checked and is complete and correct.

Changes to manuscript: None.

Additional Editor Comments:

Additional editor comment #1: Please use systematically the more accurate term to define the particular form of xerosis cutis you are speaking about (senile xerosis, drug induced xerosis, general skin dryness, etc.); without this precision in mind, miss-interpretations of the data could occur. For example, line 214, § “Ceramide-EOS, -NH and –EOH, the term xerosis cutis is not enough precise. May be a paragraph specifically dedicated to the description of xerosis cutis forms would be informative.

Response: Thank you very much. We added a paragraph describing different forms of xerosis cutis related to this manuscript. Following the description, we updated the terms including your provided example in the results and discussion sections of the manuscript.

Changes to manuscript: Different forms of xerosis cutis were investigated. Among the included articles, five examined elderly participants whose dry skin conditions were indicated either to be associated with aging [26] or as senile xerosis [25, 28, 33, 38] where especially older people had dry skin. Here, we represented this condition as ‘senile xerosis’. Skin dryness of persons with diabetes is described as diabetic xerosis which may be considered as one particular form of xerosis cutis. One study, which investigated dry skin in cancer patients whose skin dryness was induced by oral intake of erlotinib drug, is reported as drug-induced xerosis [43]. Two studies analyzed markers in the dry skin of patients undergoing hemodialysis [23, 29]. In all other articles, where studies were conducted on apparently healthy participants (not mentioning any underlying internal condition), the subject’s skin dryness was referred to as ‘general skin dryness’. (Study characteristics, page 9, line 173 to 182)

Following terms were updated in the results and discussion sections of the manuscript:

- ‘in one separate study, which investigated the amount of total ceramide in cancer patients whose dry skin was induced by oral intake of erlotinib drug’ was deleted and ‘drug-induced xerosis’ was added. (page 19, line 199). 

-‘general dry skin condition (conducted on healthy participants; e.g., studies which did not mention any internal or systemic disease of the subjects)’ was replaced by ‘general skin dryness’ (page 19, line 201). 

-‘general dry skin condition’ was replaced by ‘general skin dryness’ (page 19, line 209). 

- ‘aged xerotic skin’ was replaced by ‘senile xerosis’ (page 19, line 210).

-‘dry skin’ was replaced by ‘general skin dryness’ (page 19, line 212). 

-‘erlotinib induced xerosis’ was replaced by ‘drug-induced xerosis’ (page 19, line 212). 

- ‘aged xerotic subjects’ was replaced by ‘subjects with senile xerosis’ (page 20, line 216).

-‘general dry skin’ was replaced by ‘general skin dryness’ (page 20, line 217). 

- ‘skin dryness’ was replaced by ‘dry skin’ (page 20, line 218). 

-‘erlotinib induced xerosis’ was replaced by ‘drug-induced xerosis’ (page 20, line 220). 

-‘xerosis cutis’ was replaced by ‘general skin dryness’ (page 20, line 226). 

-‘erlotinib induced xerosis’ was replaced by ‘drug-induced xerosis’ (page 20, line 232). 

-‘general dry skin’ was replaced by ‘general skin dryness’ (page 21, line 239). 

- ‘skin dryness’ was replaced by ‘different dry skin conditions’ (page 21, line 247).

-‘dry skin’ was replaced by ‘drug-induced xerosis’ (page 21, line 255). 

-‘general dry skin’ was replaced by ‘general skin dryness’ (page 22, line 261). 

-‘erlotinib induced xerosis’ was replaced by ‘drug-induced xerosis’ (page 22, line 268). 

- ‘xerosis cutis’ was replaced by ‘dry skin’ (page 22, line 269). 

-‘general dry skin’ was replaced by ‘general skin dryness’ (page 22, line 274). 

- ‘xerosis cutis’ was replaced by ‘senile xerosis’ (page 22, line 277).

- ‘aged dry skin’ was replaced by ‘senile xerosis’ (page 23, line 282).

-‘general dry skin’ was replaced by ‘general skin dryness’ (page 23, line 293). 

- ‘xerosis cutis with underlying conditions’ was replaced by ‘senile xerosis and diabetic xerosis’ (page 23, line 299).

-‘general dry skin’ was replaced by ‘general skin dryness’ (page 24, line 304, 309, 311, 322). 

- ‘the dry skin of aged subjects’ was replaced by ‘senile xerosis’ (page 25, line 326).

-‘general dry skin’ was replaced by ‘general skin dryness’ (page 25, line 328, 332). 

- ‘the dry skin of aged people’ was replaced by ‘senile xerosis’ (page 26, line 351).

- ‘in the (dry) skin of diabetic patients’ was replaced by ‘diabetic xerosis’ (page 27, line 371, 392).

- ‘scalp skin’ was changed to ‘scalp skin (general skin dryness)’ (page 27, line 380, 385).

-‘general dry skin’ was replaced by ‘general skin dryness’ (page 33, line 420, 428). 

Following information was updated in S2 Appendix of the manuscript:

- ‘Age range: years’ was replaced by ‘Age range: 30 to 68 years’ (page 13). This information was missing during the first submission of S2 Appendix. We have found and fixed the error.

Additional editor comment #2: It would probably be better to use the term “carboxylic acids” (plural) when a particular carboxylic acid is not looked for, e.g., line 309.

Response: We replaced the term “carboxylic acid” with “carboxylic acids” in line 288 and line 325.

Changes to manuscript: Twenty-five NMFs components were reported in different dry skin etiologies, which include most standard amino acids, ornithin, citrulline, gamma-aminobutyric acid, urocanic acid, carboxylic acids and pyrrolidone carboxylic acid. (Page 23, line 288). … Urocanic acid, carboxylic acids and pyrrolidone carboxylic acid (PCA) (Page 24, line 325).

Additional editor comment #3: line 286, please change “the is” to ‘there is”.

Response: We have corrected the typing mistake.

Changes to manuscript: However there is at least one study which found either ‘unclear’ or ‘no’ association of these amino acids with general skin dryness [27]. (page 23, line 300)

Reviewer's comments:

Reviewer #1:

Comment #1: Is the manuscript technically sound, and do the data support the conclusions?

Response: We are grateful to know that the Reviewer's Responses to the Question was ‘Yes’.

Changes to manuscript: N/A.

Comment #2: Has the statistical analysis been performed appropriately and rigorously?

Response: The Reviewer's Responses to the Question was ‘N/A’ which aligns with the descriptive nature of our review.

Changes to manuscript: N/A.

Comment #3: Have the authors made all data underlying the findings in their manuscript fully available?

Response: The Reviewer's Responses to the Question was ‘No’. We checked all the documents again and it seems that, we provided all relevant data in the manuscript and the supporting documents (S1 Appendix to S4 Appendix). 

Changes to manuscript: None.

Comment #4: Is the manuscript presented in an intelligible fashion and written in standard English?

Response: We are grateful to know that the Reviewer's Responses to the Question was ‘Yes’.

Changes to manuscript: N/A.

Comment #5: This is an interesting systematic review. I think it would be useful to include the keywords used in the search in the main paper as this is key. In general legs are drier than forearms. It might be interesting to comment in the discussion on any differences observed between studies where leg skin was used versus arm skin. For completion, the PROSPERO details should be included in the supplementary data.

Response: Thank you very much for your encouraging comments and insightful suggestions. Now, we provided additional keywords like ‘ceramide’, ‘cytokine’, ‘enzyme’ from the initial search. Statements were added in the discussion section about differences between leg and arm skin. 

Our manuscript is structured according to the PRISMA reporting guideline. Therefore, we feel that relevant information is provided. In addition, we provided the reference to the PROSPERO protocol and provided a copy of the protocol as supporting information as indicated in PLOS ONE manuscript submission guidelines.

Changes to manuscript: The key words ‘ceramide’, ‘cytokine’, ‘enzyme’ were added. This paragraph was added in the discussion section: “It is also well known that the occurrence and severity of xerosis cutis is skin area specific, for example, in senile xerosis the legs are drier than the arms [2]. However, the substantial heterogeneity of the reviewed evidences makes these intended comparisons almost impossible. In addition, we did not include any study that compared skin dryness or markers from both the arms and leg skin areas.” (page 35, line 467 to 471)

---

## [Editor Report · Decision Letter 1]

29 Nov 2021

Molecular characterization of xerosis cutis: a systematic review

PONE-D-21-19523R1

Dear Dr. Amin,

We’re pleased to inform you that your manuscript has been judged scientifically suitable for publication and will be formally accepted for publication once it meets all outstanding technical requirements.

Kind regards,

Michel Simon, Ph. D.

Academic Editor

PLOS ONE

Additional Editor Comments (optional):

The Authors satisfactorily answered the last questions of Reviewer and Academic Editor, and followed their modification requirements.
---

## [Editor Report · Acceptance letter]

2 Dec 2021

PONE-D-21-19523R1 

Molecular characterization of xerosis cutis: a systematic review 

Dear Dr. Amin:

I'm pleased to inform you that your manuscript has been deemed suitable for publication in PLOS ONE. Congratulations! Your manuscript is now with our production department. 

Kind regards, 

on behalf of

Dr. Michel Simon 

Academic Editor

PLOS ONE